PLOS · Biology

META-RESEARCH ARTICLE

# Testing the reproducibility of ecological studies on insect behavior in a multi-laboratory setting identifies opportunities for improving experimental rigor

Carolin Mundinger[1☉*], Nora K. E. Schulz[2☉], Pragya Singh[3☉], Steven Janz[3], Maximilian Schurig[2], Jacob Seidemann[4], Joachim Kurtz[2,5], Caroline Müller[3,5], Holger Schielzeth[4,5], Vanessa T. von Kortzfleisch[1], S. Helene Richter[1,5*,]

1 Department of Behavioural Biology, University of Münster, Münster, Germany, 2 Institute for Evolution and Biodiversity, University of Münster, Münster, Germany, 3 Department of Chemical Ecology, Bielefeld University, Bielefeld, Germany, 4 Institute of Ecology and Evolution, Friedrich Schiller University Jena, Jena, Germany, 5 Joint Institute for Individualisation in a Changing Environment, University of Münster and Bielefeld University, Münster and Bielefeld, Germany

☉ These authors contributed equally to this work.
* richterh@uni-muenster.de (SHR); c.mundinger@uni-muenster.de (CM)

## Abstract

The reproducibility of studies involving insect species is an underexplored area in the broader discussion about poor reproducibility in science. Our study addresses this gap by conducting a systematic multi-laboratory investigation into the reproducibility of ecological studies on insect behavior. We implemented a 3 × 3 experimental design, incorporating three study sites, and three independent experiments on three insect species from different orders: the turnip sawfly (*Athalia rosae*, Hymenoptera), the meadow grasshopper (*Pseudochorthippus parallelus*, Orthoptera), and the red flour beetle (*Tribolium castaneum*, Coleoptera). Using random-effect meta-analysis, we compared the consistency and accuracy of treatment effects on insect behavioral traits across replicate experiments. We successfully reproduced the overall statistical treatment effect in 83% of the replicate experiments, but overall effect size replication was achieved in only 66% of the replicates. Thus, though demonstrating sufficient reproducibility in some measures, this study also provides the first experimental evidence for cases of poor reproducibility in insect experiments. Our findings further show that reasons causing poor reproducibility established in rodent research also hold for other study organisms and research questions. We believe that a rethinking of current best practices is required to face reproducibility issues in insect studies but also across disciplines. Specifically, we advocate for adopting open research practices and the implementation of methodological strategies that reduce bias and problems arising from over-standardization. With respect to the latter, the introduction of systematic variation through multi-laboratory or heterogenized designs may contribute to improved reproducibility in studies involving any living organisms.

**Data availability statement:** The code and data for all modeling are available at Zenodo https://zenodo.org/records/14002690.

**Funding:** This research was funded by the German Research Foundation (DFG) as part of the CRC TRR 212 (NC³) – Project numbers 316099922 (CRC overall), 396776123 (SHR), 396777467 (CM), 396780003 (JK), and 396776775 (HS). The funders had no role in study design, data collection and analysis, decision to publish, or preparation of the manuscript.

**Competing interests:** The authors have declared that no competing interests exist.

**Abbreviations:** CI, confidence interval; DMD, 4,8-dimethyl decanal; EDA, The Experimental Design Assistant; GLMM, generalized linear mixed models; OLRE, observation-level random effect; PCI, post-contact immobility.

## 1. Introduction

Reproducibility, i.e., the ability of a result to be replicated by an independent experiment in the same or different laboratory [1]; also referred to as replicability [2], is a cornerstone of any scientific method. Results that cannot be independently reproduced cause scientific uncertainty, hinder scientific progress, and incur costs to science and society. Attempts to replicate scientific findings have produced mixed to rather discouraging results [3–5], raising concerns whether reproducibility is as high as is desirable and coining the term "reproducibility crisis" (e.g., [3,5–7]). Furthermore, more than 70% of researchers who responded to a survey reported having tried and failed to reproduce another scientist's experiments, and more than half admitted having failed to reproduce their own experiments [5]. Interestingly, this crisis does not seem to be restricted to a specific research discipline but may affect disciplines as diverse as human medicine [8], psychology [6,9], and economics [10,11].

Historically, the discussion about poor reproducibility was sparked by a multi-laboratory study on mouse phenotyping [12]. In this landmark study, eight different mouse strains were simultaneously investigated in a battery of six behavioral paradigms at three different sites. Although the test set-up and the environmental conditions were rigorously standardized across the three laboratories, Crabbe and colleagues (1999) detected strikingly different results in the three labs, with some behavioral tests even yielding contradictory findings. The authors therefore concluded that "experiments characterizing mutants may yield results that are idiosyncratic to a particular laboratory" [12].

In subsequent years, a series of single- and multi-laboratory as well as systematic replication studies supported these findings, uncovering further problems with reproducing previous findings (e.g., [13–16]). Consequently, researchers started to search for causes of poor reproducibility and ways out of the crisis. Along these lines, diverse threats to reproducibility have been proposed, with a lack of scientific rigor, low statistical power, publication bias, and analytical flexibility being among the most discussed ones (e.g., [17–21]). To solve these issues, a number of strategies have been developed that aim to overcome biases in the experimental design (e.g., The Experimental Design Assistant—EDA [22]) and/or to improve reporting of the study setup (e.g., ARRIVE guidelines [23,24]). Likewise, the pre-registration of studies as well as the publication of registered reports have been emphasized to address biases at the publication level (e.g., [25,26]).

When considering animal research in particular, yet another problem has been identified, namely the systematic neglect of biological variation (e.g., [1,27–30]). The underlying idea here is that the response of an animal to an experimental treatment depends not only on the properties of the treatment but is a product of the animal's genotype, parental effects, and its past and present environmental conditions (see also "reaction norm perspective" [31–33]). As laboratory experiments are usually conducted under highly standardized conditions, only a very narrow range of environmental conditions is represented, thereby limiting the inference space of the whole study. Efforts taken to increase reproducibility (i.e., through rigorous

standardization) therefore compromise external validity, simply because they restrict the range of environmental conditions to a specific "local set". This apparent increase in reproducibility at the expense of external validity has repeatedly been highlighted as "standardization fallacy" [34,35] and explains why results can differ when experiments are replicated. Interestingly, however, this reasoning has been almost exclusively used to explain poor reproducibility of preclinical animal research with laboratory rodents. This is particularly surprising as the underlying logic should apply to all living organisms and might explain why discrepant findings arise, whenever animals are studied under standardized laboratory conditions, independent of the study species, the experimental purpose, or the research area.

We here explore if similar problems arise in experimental studies on insects. The use of insects for laboratory experiments is common in many disciplines, but—despite the multitude and diversity of research approaches—has attracted far less attention in terms of reproducibility. To our knowledge, there have been no systematic (meta-) research projects on reproducibility of insect studies to date. Moreover, we know only little about the extent to which the reproducibility crisis plagues the community of studies in behavior, ecology, and evolution, although it appears that the field is not impervious to reproducibility issues (e.g., [36–38]).

The present study aimed to bridge this gap, by systematically exploring the reproducibility of simple ecological studies on insect behavior. In a multi-laboratory approach, we conducted three experiments at three locations with three different study species, each following the same standardized protocol. The study species included the turnip sawfly *Athalia rosae* (Hymenoptera), the meadow grasshopper *Pseudochorthippus parallelus* (Orthoptera), and the red flour beetle *Tribolium castaneum* (Coleoptera), thus representing three frequently studied insect species across three different orders. Each experiment had already been conducted in a previous study in a single laboratory, ensuring that the treatments had produced significant effects in the past (see for *A. rosae*: [39]; for *P. parallelus*: [40]; for *T. castaneum:* [41]). This implies that one laboratory had extensive experience with one of the study species and the experiment, while the other two laboratories were inexperienced and relied on protocols. Consequently, we expected a higher reproducibility of treatment effects in laboratories that had already conducted the respective pilot studies compared to inexperienced laboratories new to both the respective study organism and the experimental set-up. Furthermore, we predicted that manual handling during testing would introduce more between-laboratory variation than assays relying on observation alone.

## 2. Animals, materials, and methods

Using a multi-laboratory approach, we investigated the reproducibility of three experiments involving three insect species: the turnip sawfly (*A. rosae*), the meadow grasshopper (*P. parallelus*), and the red flour beetle (*T. castaneum*). These organisms exemplify three types of model systems. Individuals of *P. parallelus* were collected from the wild for this study. In contrast, *T. castaneum* serves as a classic example of a laboratory-adapted insect model, having been bred exclusively in laboratory conditions for over a decade [42]. The species *A. rosae* represents an intermediate state, as its populations originated from a laboratory culture that was annually supplemented with wild-caught individuals. Each experiment was designed by one of the participating laboratories to address specific evolutionary and ecological questions relevant to the study species (Fig 1).

In the first experiment, we examined the effects of starvation on larval behavior in *A. rosae*. Starvation, a common ecological stressor, can profoundly affect behavioral responses. In this experiment, we specifically measured post-contact immobility (PCI) and activity following a simulated attack. Based on previous findings [39], we hypothesized that, compared to non-starved larvae, starved larvae would exhibit shorter PCI durations and increased activity levels, an adaptive strategy to enhance foraging success and food location under nutritional stress. The PCI quantifications required manual handling of the larva, whereas evaluating activity required little human intervention and handling. A comparison of these two tests presents an opportunity to investigate which behavioral test is more prone to experimental variability introduced by handling. We predicted higher between-laboratory variation in results of the PCI trait compared to results on activity.

| Laboratory of origin of respective experiment | | |
|---|---|---|
| **Laboratory** | Lab A (Bielefeld) | Lab B (Jena) | Lab C (Muenster) |
| **Model organism** | *Athalia rosae* | *Pseudo-chorthippus parallelus* | *Tribolium castaneum* |
| **Research question** **(Measurement)** | Effects of starvation (PCI, distance moved) | Substrate choice (Choice of colored underground) | Niche preference (Choice of flour) |

**Fig 1. Overview of the 3 × 3 design with the participating laboratories, study species, and research questions of the respective experiments.** PCI: post-contact immobility. Photo credit: Athalia: Pragya Singh; Pseudochorthippus: Holger Schielzeth; Tribolium: Tobias Prueser.

In the second experiment, we investigated the relevance of color polymorphism for substrate choice in *P. parallelus*. The experiment used two color morphs (green and a brown) as the statistical treatment effect of interest to test for morph-dependent microhabitat choice and crypsis [40]. We assessed the preference of grasshoppers for green or brown patches to evaluate whether color morphs selectively choose backgrounds (= substrates) that match their own color and thus enhance their camouflage. We predicted that each morph would preferentially select a substrate that matches its body color.

The third experiment focused on *T. castaneum*, where we assessed niche preference by offering beetles a choice between flour types conditioned by beetles with or without functional stink glands [41,43]. In this species, adult beetles secrete quinone-based compounds that not only alter the microflora [44] and provide external immunity [45] but also serve as signals of population density [46]. These secretions can create microhabitats of varying quality, potentially guiding beetles in selecting optimal habitats. We predicted that larvae and adult beetles differed in their niche choice, with larvae showing a preference for conditioned flour containing antimicrobial secretions, while adults avoid this conditioned flour.

To ensure consistency across the participating laboratories, each experiment followed a standardized protocol. Behavioral assays were conducted using predefined methodologies to minimize between-laboratory variation. Data collected from the three laboratories were then analyzed to assess the reproducibility of the experimental results. Environmental conditions, such as temperature, humidity, and light cycles were controlled and kept as consistent as possible across all laboratories (see S1–S3 Tables). Diets were standardized to a large degree, but were not completely identical, as each laboratory procured the food for the insects itself by either buying, growing, or collecting the necessary dietary component. In detail, cabbage for *A. rosae* and grass blades for *P. parallelu*s were freshly sourced locally. For *T. castaneum*, organic wheat flour type 550 with 5% brewer's yeast was used. While the yeast came from the same batch, flour was bought from different distributors. All details can be found in the supplement table listing laboratory-specific conditions (see S1–S3 Tables).

### 2.1. Effect of starvation on behavior of *Athalia rosae*

**2.1.1. Study organism.** We investigated the effects of starvation on PCI and activity levels in larvae of *A. rosae*. The larvae of the species feed on various Brassicaceae plants and can be an agricultural pest [47]. In the larval stage, individuals can swiftly defoliate their host plants, consequently facing periods of starvation [48]. Besides finding food, larvae also need to manage predation hazards. PCI is a behavioral response to physical interaction with a predator. During PCI individuals remain motionless for a certain duration [39], a phenomenon that is also referred to as post-predation immobility, tonic immobility, thanatosis, or "death-feigning" behavior [49]. PCI is hypothesized to function as an antipredator tactic [49] and is also used as a proxy for boldness or risk-taking behavior in animals [50,51]. We chose the behavioral traits of PCI and activity because we previously documented a pronounced starvation effect on these traits, with starved larvae exhibiting shorter PCI duration and higher activity levels [39].

The sawflies used in this study originated from the Bielefeld laboratory stock population, initially established from adults collected in the vicinity of Bielefeld, Germany. This stock population was annually supplemented with wild-caught *A. rosae* adults. Stock population sawflies were housed in mesh cages (60 × 60 × 60 cm) within a laboratory environment at a 16:8-h light-to-dark cycle, approximately 60% relative humidity, and room temperature (15–25°C). For this study, multiple male and female adults were introduced into a cage with mustard (*Sinapis alba,* Brassicaceae) plants for oviposition, giving rise to male and female offspring. After 1 week, the newly hatched larvae were reared on non-flowering plants of cabbage (*Brassica rapa* var. *pekinensis*, Brassicaceae). Mustard and cabbage had been cultivated from seeds in a climate chamber and a greenhouse, respectively. Third- and fourth-instar larvae were collected from the cage, put individually in Petri dishes (5.5 cm diameter) lined with slightly moistened filter paper and provided with cabbage leaf discs, and were sent via mail to all three laboratories (70 larvae to each laboratory). Upon arrival, the larvae were transferred to fresh Petri dishes and were given *ad libitum* access to cabbage leaf discs obtained locally. Some larvae molted during the experimental assay and were excluded from the experiment. The experiment was performed within 4 days after the arrival of the larvae.

**2.1.2. Larval behavioral traits: PCI assay and activity.** For the experiment, the larvae were moved to clean Petri dishes with moist filter paper (1 larva per dish) and randomly allocated to a control or starvation treatment (*N* = 30 per treatment). In the starvation treatment larvae had no access to cabbage leaves, while in the control treatment, larvae were provided cabbage leaves *ad libitum* (see S1 Fig). The position of the Petri dishes of the two treatments was alternated to avoid any spatial effects. After 3 h of treatment, the PCI duration of all larvae was measured. For measuring PCI, a clean Petri dish (5.5 cm diameter) was prepared and a graph-paper was placed beneath the Petri dish. To induce PCI, a larva was grasped in the middle of the body using soft-tip spring-steel forceps and dropped from a height of 5 cm onto the clean Petri dish. The larva was considered to exhibit PCI if it curled up its body and stayed motionless in this curled-up posture for at least one second. If the larva did not show PCI, the forceps stimulation was repeated up to two more times. The PCI duration was measured starting from the time the larva showed PCI until the time it uncurled its body and moved at least 1 cm. Each larva was monitored for a maximum of 10 min. After the PCI assay, all larvae were returned to their Petri dishes. One hour after measuring PCI (= after 4 h of starvation treatment), the activity level of each larva was measured. Larvae were moved individually to clean, empty Petri dishes (5.5 cm diameter), and their behavior was recorded and tracked for 10 min using a video camera. Six Petri dishes were recorded in parallel. Using a tracking software (different software used depending on laboratory, see S1 Table), the data on the distance moved for each individual was extracted from the videos.

### 2.2. Substrate choice in *Pseudochorthippus parallelus*

**2.2.1. Study organism.** *Pseudochorthippus parallelus* is a species of grasshopper that is widespread across most of Europe and occurs in multiple discrete color variants that co-occur in local populations [52]. This raises the evolutionary

question of why and how this phenotypic polymorphism is maintained within populations. It has been hypothesized that a trade-off between crypsis and thermoregulation results in balancing selection and thus a maintenance of color-morph diversity [53]. Experimental evidence suggests that in the laboratory, greener individuals have a stronger preference to perch on green backgrounds than brown individuals [40], supporting the hypothesis of morph-differential microhabitat choice. The original study had been conducted with three color morphs and we simplified the experiment to the two most extreme morphs (uniform green and uniform brown) for the purpose of our multi-laboratory replication study.

**2.2.2. Substrate choice in green and brown morphs.** Experimental individuals were wild-caught from a grassland near Jena, Germany. Two days after the mature grasshoppers were caught, they were transported in small vials by train to the other laboratories, with experimental animal for Jena also being transported by train for about the same duration of time. After arrival, experimental animals were stored in refrigerators overnight. The next day, we transferred individuals to experimental plastic cages (37 × 22 × 24.5 cm) with one individual per cage. The floor of cages was covered by a laminated sheet which contained a 2 × 2 checkerboard pattern of brown and green patches (see S2 Fig). As a modification to the original experiment [40], brown and green patches were matched for brightness. Experimental individuals were supplied with freshly cut grass leaves potted in small water-filled vials placed at the center of the cage and female were also provided with a small sand pot for egg laying. Cages, vials, and lining for the floor were provided by the Jena laboratory, while food was collected locally and consisted of blades of wild-growing Poaceae, mainly *Dactylis glomerata.* After the last measurement of each day, food was replaced if needed.

We arranged cages in a way that individuals of the same sex, but different morphs were next to each other and distributed these paired cages randomly across the room. After one day of acclimatization, the assay started. To record substrate choice, we manually documented the color of the patch that each individual was sitting on (using the position of the head if the individual was sitting on the border of two patches, as in [40]). Observations were repeated on 10 consecutive days, with five measurements taken per day. All measurements were performed between 9 am and 4 pm, to ensure sufficient light and relatively warm temperatures, and at least 1 h elapsed between recordings. Data from individuals that were not sitting on the floor of the cage were excluded from the analysis.

### 2.3. Niche preference in *Tribolium castaneum*

**2.3.1. Study organism.** Red flour beetles are group-living insects and a global pest to stored grain. The beetles live in overlapping generations within their food source. Adults provide external immunity via quinone-rich stink gland secretions to their offspring and conspecifics [54–56]. However, quinones in high concentrations can have toxic effects [57]. Red flour beetles can exhibit cannibalistic behavior, wherein larvae and mainly female adults feed on eggs and pupae [58]. Therefore, individuals face a trade-off in their niche choice. Due to their high feeding rate, which enables rapid growth, larvae are especially prone to oral infections by entomopathogens. However, only mature adults can produce protective stink gland secretions, which implies that larvae given a choice should prefer flour-containing stink gland secretions. A previous choice assay supported this prediction by demonstrating larval preference for conditioned flour containing antimicrobial secretions [41]. In contrast, virgin females showed a preference for flour conditioned by beetles without functional stink glands and thus drastically reduced secretions [41]. Females might choose this niche because it signals a lower population density, and thus a lower risk of cannibalism. Furthermore, the absence of quinones allows higher concentrations of the aggregation pheromone 4,8-dimethyl decanal (DMD) to persist [59], which attracts adult beetles.

The animals used in this study were from the CRO1 strain [42], which is derived from originally 165 mating pairs caught in Croatia in 2010. The population has been kept in the laboratory in non-overlapping generations on organic wheat flour (DM, type 550) with 5% brewer's yeast. The substrate was heat sterilized at 75°C and they were kept at 30°C and 70% humidity at a 12 h/12 h light/dark cycle.

The experiment was carried out using virgin adult females and larvae of unknown sex. The Münster laboratory provided a parental population of beetles to each laboratory, along with RNAi-treated knockdown and control beetles, which were

used to condition the flour [41]. Beetles that receive a knockdown for the *Drak* gene during the pupal stage do not develop functioning stink glands, therefore flour conditioned by them hardly contains any of the characteristic quinone-containing secretions. The flour conditioned this way is referred to as "*Drak* flour" below. The second flour type was conditioned by beetles from the control group, which received a *Gfp* RNAi control injection and produced normal levels of stink gland secretions. The successful knockdown was confirmed via photometric measurements of quinones at specific wavelengths. Beetles were sent via mail to all three laboratories.

**2.3.2. Niche choice assay in larvae and adults.** In the behavioral assay, we gave the beetles the choice between two different flour types, either containing quinone-rich stink gland secretions or having these drastically reduced. We conducted the assay in Münster and Bielefeld in late spring of 2023, and in Jena in May 2024. At the time of the assay, larvae were 17 days old and virgin females were 40–45 days. The sex was determined during the pupal stage, and females were separated from males to ensure their virgin status.

For the behavioral assay, we prepared Petri dishes (9 cm diameter) by marking a center line and spreading 0.3 g of each flour type on either side, leaving a 1 cm section in the middle blank (see S3 Fig). The order of the Petri dishes regarding side orientation was randomized and the observer was blind to the identity of the different flour types. We conducted the experiment in two blocks separated by two weeks. Each block had a sample size of *N* = 40 or 60 Petri dishes per life stage. Three virgin female adults or three larvae were used per Petri dish. Conducting the assay with three individuals reduced the amount of missing data because individual arenas could still be scored, even if one or two individuals had fallen on the back and could not make an active choice. Also, multiple adults in the same Petri dish can flip each other back on their feet through random interactions. At the start of the experiment, animals were placed with soft-tip spring-steel forceps in parallel orientation along the central line of the dish at the distal end of the flour. We took the first measurement 30 min after all animals were put into the arenas. For 6 h, we then recorded every 30 min the position of each individual in each Petri dish, and for the adults whether it was on its feet or laying on the back. One additional measurement was taken 24 h after the start. Only data from individuals on their feet were used in the analysis. Thus, some of the individuals did not produce data for all 14 time points.

## 2.4. Statistical analysis

All analyses were done in R version 4.3.2 [60]. The code and data for all modeling are available at Zenodo (https://zenodo.org/records/14002690). Data were analyzed using univariate generalized linear mixed models (GLMM; package "lme4" version 1.1-35.1 [61] and package "lmerTest" version 3.1-3 [62]). Model residuals were inspected using the DHARMa package version 0.4.6 [63].

For *A. rosae*, we modeled the duration of PCI and the distance moved. Variables were appropriately transformed to improve the distribution and variance homogeneity of model residuals (y' = log (y + 1) for PCI duration and y' = sqrt(y) for distance moved). Both transformed variables were modeled using Gaussian error distributions. For *P. parallelus,* the choice was modeled as the probability of an individual sitting on a green patch in the form of a binary response using binomial models with logit link function. As we did not find any difference in patch preference between the two sexes in *P. parallelus* (GLMM with the structure green preference ~ morph + sex + (1|lab) + (1|id); β = 0.05 ± 0.11, z = 0.40, *p* = 0.69; estimate refers to difference in patch choice of males relative to females; see also S4 Fig), we excluded sex as a factor from the analysis to keep the models aligned with those of *A. rosae* and *T. castaneum*. For *T. castaneum*, beetles were always recorded as three indistinguishable individuals grouped on a single Petri dish. Therefore, we calculated the preference as the number of individuals found in quinone-reduced flour relative to the number of individuals in control flour. The response was thus modeled as proportion data using binomial models with logit link function. For an overview of all biological variables modeled, see Table 1.

*P. parallelus* models fitted "*individual identity*" and *T. castaneum* models fitted "*Petri dish identity*" as random effects, to account for the non-independence of repeated measures. To cope with overdispersion in count data, we further added an

**Table 1. Description as well as sample sizes of the data for the mixed models.** Sample sizes are the total number of units/individuals recorded across all three laboratories. OLRE is the observation-level random effect. For detailed sample sizes see S4–S6 Tables.

| Experiment | Outcome measure | Description and scale before transformation | Modelled response and distribution | Treatment contrast (fixed effects) | Random effects | Total sample size across all labs |
|---|---|---|---|---|---|---|
| *Athalia rosae* | Post-contact immobility (PCI) | Duration of PCI: (0–600 s) | Log + 1 (gaussian) | **Starvation**: Starved/control | | 180 individuals |
| *Athalia rosae* | Distance moved | Distance moved (0–144 cm) | Sqrt (gaussian) | **Starvation**: Starved/control | | 180 individuals |
| *Pseudochorthippus parallelus* | Substrate choice | 1 = on green<br>0 = on brown | Binary (binomial) | **Morph**: Green/brown | individual ID | 185 individuals |
| *Tribolium castaneum* | Flour choice | Number of individuals on quinone-containing flour to number of individuals on control flour (e.g., 3/0 or 1/2) | Proportion (binomial) | **Life stage**: Larvae/adult | dish ID + OLRE | 642 individuals, 214 dishes |

observation-level random effect (OLRE; [64]) to the *T. castaneum* models. We tested for overdispersion using the "performance" package version 0.12.0 [65].

We evaluated the reproducibility of the treatment effects across all three laboratories, by following the approach of von Kortzfleisch and colleagues (2020) [66]: Firstly, we estimated the consistency of treatment effects across the laboratories. Secondly, we evaluated how accurately each laboratory was able to predict the overall effect and effect size.

To evaluate the consistency of treatment effects, we fitted GLMMs with only the treatment as fixed effect separately for each of the experiments (plus random effects for *P. parallelus* and *T. castaneum* as described above). In a second analysis, the data were pooled across experiments and "*laboratory*" as well as the interaction between "*treatment*" and "*laboratory*" were fitted as fixed effects. We then used likelihood ratio tests (car-package version 3.1-2 [67]) to assess the statistical significances of the "*treatment-by-laboratory*" interaction terms.

For the evaluation of how accurately the laboratories were able to predict the overall treatment effect, three measurements were calculated: the coverage probability ($P_{cov}$), the proportion of consistently significant results ($P_{sig}$), and proportion of accurate results ($P_{acc}$). $P_{cov}$ represents the proportion of experiments in which the 95% confidence interval ($CI_{95}$) of the estimate covered the meta-analytic average effect. The meta-analytic overall effect size was estimated by a random-effect meta-analysis using the metafor-package (version 4.4-0 [68]) based on the individual treatment effect sizes and standard errors of all laboratories.

The proportion of consistent significant results ($P_{sig}$) quantifies the proportion of laboratory-specific experiments that are congruent with the overall significance ($P < 0.05$ versus $P > 0.05$) and, if significant, also in sign. We did not find cases of significance in opposite directions, so do not discuss this situation further. To determine overall statistical significance, we examined whether the $CI_{95}$ of the overall effect as estimated by meta-analyses (as described above) contained 0 (i.e., indicating an overall not significant effect) or not (i.e., indicating an overall significant effect).

For the most conservative measure, the proportion of accurate results ($P_{acc}$), we then combined both measures of $P_{cov}$ and $P_{sig}$. To evaluate whether a laboratory accurately predicted the overall effect and effect size, two conditions needed to be met: the 95% confidence interval ($CI_{95}$) of the estimates covered the meta-analytic average effect (see $P_{cov}$), and the statistical significance had to be in the same direction as to the overall effect (see $P_{sig}$; compare [69] and [66] for graphical illustrations of this notion).

## 3. Results

### 3.1. Experiment-specific results

**3.1.1. Activity of *Athalia rosae*.** We examined the effects of starvation on larval behavior in *A. rosae*. Overall, we recorded 360 measurements across the two behavioral activity measures of 180 individual sawfly larvae. We found a

significant correlation between the two activity measurements: the longer the distance moved, the shorter the PCI duration ($\rho$ = −0.57, $p$ < 0.001).

Across laboratories, we found treatment-specific differences for both behaviors (PCI and distance moved). Starved *A. rosae* larvae remained significantly shorter in PCI (two-sided Wilcoxon signed-rank test, $W$ = 6,144, $p$ < 0.001; see Fig 2A) and were more active in the form of a greater distance moved ($W$ = 1,475, $p$ < 0.001; see Fig 2C).

When comparing the results for each of the laboratories, we found that the overall effect of starvation on PCI duration was reproduced in only two out of the three replicates (Lab A: $W$ = 871.5, $p$ < 0.001; Lab C: $W$ = 621, $p$ = 0.01; see Fig 2B). While the direction of the effect was similar for those experiments, the length of the PCI duration of the control group was much more prolonged and less variable in Lab A (on average 425 ± 182 s) compared to Lab C (200 ± 213 s). Only

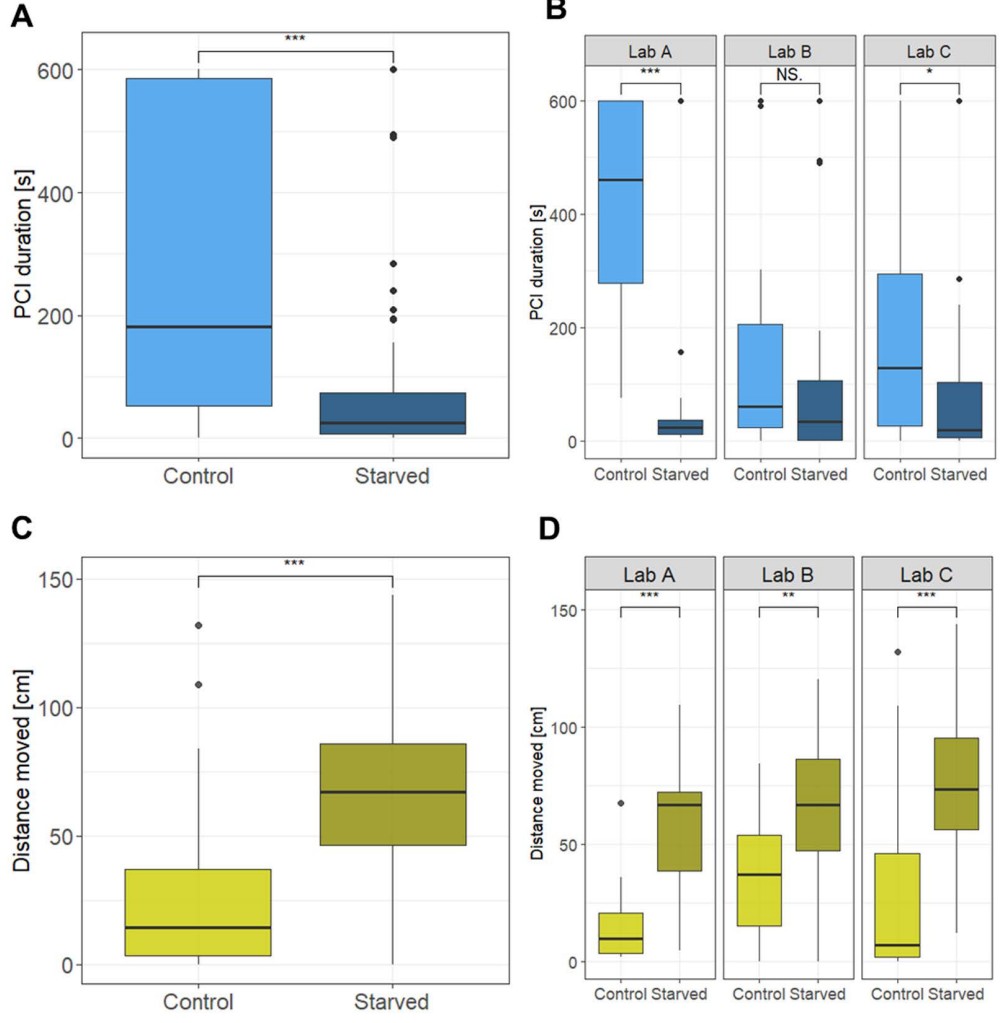

**Fig 2. Effect of treatment (control or starvation) on the post-contact immobility (PCI) duration (A) across laboratories and (B) within laboratories and on the distance moved (C) across laboratories and (D) within laboratories in *A. rosae*.** Data are presented as box plots showing medians, 25% and 75% percentiles (lower and upper box), and 5% and 95% percentiles (lower and upper line). Statistics: Wilcoxon signed-rank test, two-sided, *$p \le 0.05$, **$p \le 0.01$, ***$p \le 0.001$. Model results from the GLMM on the transformed data show the same direction of significant differences (see Table 2). Raw data and code needed to reproduce this Figure can be found in https://zenodo.org/records/14002690, the summary data presented in the Figure is additionally listed in Table 2.

Lab C covered the overall effect size, while Lab A overestimated it (see Fig 3A). In contrast, the third laboratory did not detect a statistically significant difference between the starved and control larvae (Lab B: $W = 549.5$, $p = 0.14$; see Fig 3A; but please see critical discussion on the use of $p$-values in paragraph 4.1.3). This impaired reproducibility between laboratories was further reflected in a significant "*treatment-by-laboratory*" interaction term (LRT: $\chi^2(2) = 30.12$, $p = 0.006$).

For the distance moved, we found significant treatment differences in each of the three laboratories (Wilcoxon signed-rank test, Lab A: $W = 67$, $p < 0.001$; Lab B: $W = 239$, $p < 0.01$; Lab C: $W = 147$, $p < 0.001$; see Fig 2D). Nevertheless, we found a significant "*treatment-by-laboratory*" interaction term in the pooled analysis (LRT: $\chi^2(2) = 51.25$, $p = 0.024$). Only two laboratories (Lab A and Lab C) recovered the overall effect size, while the third laboratory (Lab B) underestimated the magnitude (see Fig 3B).

### 3.1.2. Substrate choice in *Pseudochorthippus parallelus*.

We tested whether grasshoppers' color morphs differed in their preference for either green or brown substrate patches. Overall, we recorded 8,784 positions of 185 individual grasshoppers. In 15.1% of those observations (1,329 instances), grasshoppers were sitting on the floor and could thus be assigned to one of the two patch colors (see S5 Fig). In the remaining 84.9% of observations (7,455 instances), grasshoppers were sitting on the cage walls, under the cage lids, or on the bundles of grass and were consequently not assigned any patch preference.

Among the records of grasshoppers sitting on one of the colored patches, there were no significant morph-specific differences in substrate choice, neither across all laboratories (two-sided Wilcoxon signed-rank test, $W = 3626.5$, $p = 0.33$; see Fig 4A) nor within each laboratory (Lab A: $W = 358$, $p = 0.34$, Lab B: $W = 490$, $p = 0.28$; Lab C: $W = 361.5$, $p = 0.09$;

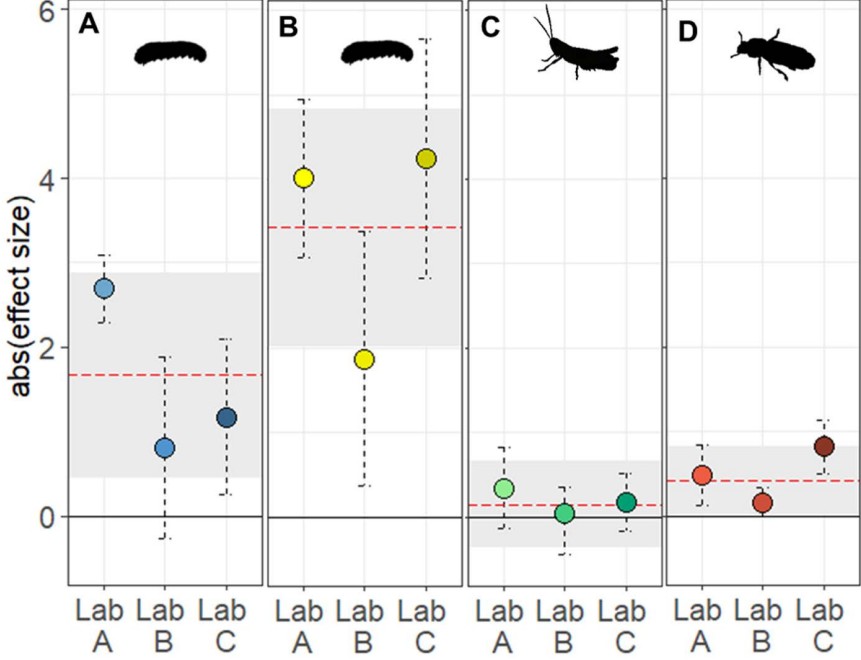

**Fig 3. Variation of treatment differences across the three replicate experiments for (A) the post-contact immobility (PCI) duration and (B) distance moved in larvae of *A. rosae*, (C) the substrate preference in the two color morphs of *P. parallelus*, and (D) the niche preference in the two life stages of *T. castaneum*.** The black solid line reflects the null effect. The red dashed line and gray area indicate the overall absolute (abs) effect size and its corresponding 95% confidence interval ($CI_{95}$). Dots and vertical dashed lines reflect the mean group differences (as estimated from the GLMMs) and corresponding $CI_{95}$ of the three laboratories. The overall absolute effect size was estimated by a random-effect meta-analysis based on the individual treatment effect sizes and standard errors of all laboratories. Raw data and code needed to reproduce this Figure can be found in https://zenodo.org/records/14002690, the summary data presented in the Figure is listed in the S4A–D Table.

**Table 2. Differences between the treatment groups of the sawfly *A. rosae*, the two morphs of the meadow grasshopper *P. parallelus,* and the two life stages of the red flour beetle *T. castaneum*, Overall effect size, $CI_{95}$ and *p*-value are estimated by the random-effect meta-analysis (in bold) as well as the individual parameters for all laboratories estimated from GLMMs. The absence of a significant difference between the two treatment groups is highlighted in gray.**

| Experiment | Outcome measure | Replicate | Effect size | SE | Upper $CI_{95}$ | Lower $CI_{95}$ | *p*-value |
|---|---|---|---|---|---|---|---|
| *Athalia rosae* | Post-contact immobility (PCI) | **Overall** | **−1.67** | **0.62** | **−0.46** | **−2.88** | **** **0.01** |
| | | Lab A | −2.70 | 0.20 | −2.29 | −3.1 | *** <0.001 |
| | | Lab B | −0.81 | 0.55 | 0.28 | −1.89 | N.S. 0.15 |
| | | Lab C | −1.17 | 0.47 | −0.25 | −2.09 | * 0.02 |
| | Distance moved | **Overall** | **3.43** | **0.72** | **4.84** | **2.01** | *** <0.001 |
| | | Lab A | 4.01 | 0.48 | 4.95 | 3.07 | *** <0.001 |
| | | Lab B | 1.87 | 0.77 | 3.39 | 0.36 | * 0.02 |
| | | Lab C | 4.25 | 0.72 | 5.66 | 2.83 | *** <0.001 |
| *Pseudochorthippus parallelus* | Substrate choice | **Overall** | **0.15** | **0.26** | **0.66** | **−0.36** | **N.S. 0.58** |
| | | Lab A | 0.34 | 0.24 | 0.82 | −0.13 | N.S. 0.16 |
| | | Lab B | −0.04 | 0.20 | 0.36 | −0.44 | N.S. 0.83 |
| | | Lab C | 0.17 | 0.17 | 0.51 | −0.18 | N.S. 0.34 |
| *Tribolium castaneum* | Niche choice | **Overall** | **−0.41** | **0.21** | **−0.00** | **−0.78** | ***** **0.05** |
| | | Lab A | −0.48 | 0.18 | −0.12 | −0.84 | ** 0.01 |
| | | Lab B | −0.16 | 0.09 | 0.01 | −0.33 | N.S. 0.06 |
| | | Lab C | −0.82 | 0.16 | −0.5 | −1.13 | *** <0.001 |

SE, standard error.

see Fig 4B). All replicate CIs covered the overall effect size but also included zero effect (see Fig 3C). The consistency of these results across the replicates with the overall effect was reflected in the non-significant "*treatment-by-laboratory*" interaction term (LRT: $\chi^2(2) = 1.55$, $p = 0.46$).

**3.1.3. Niche preference in *Tribolium castaneum*.** We tested whether adult and larval flour beetles differ in their niche preference for flour with protective quinone-rich secretions or flour with drastically reduced quinone content. Overall, we recorded 24,661 positions of *Tribolium* larvae (12,328) and adults (all virgin females, 12,333) across all Petri dishes. In 33.4% of the observations (4,116 instances), adult beetles were on their feet, so that a preference could be recorded.

We found a significantly higher preference for the secretion-less flour in adults (56.8%) compared to larvae (53.3%; two-sided Wilcoxon signed-rank test, $W = 56,790$, $p = 0.02$; see Fig 5A). When studying the preference within each laboratory, however, only two out of three laboratories replicated the differences in flour preference between the larvae and

PLOS Biology

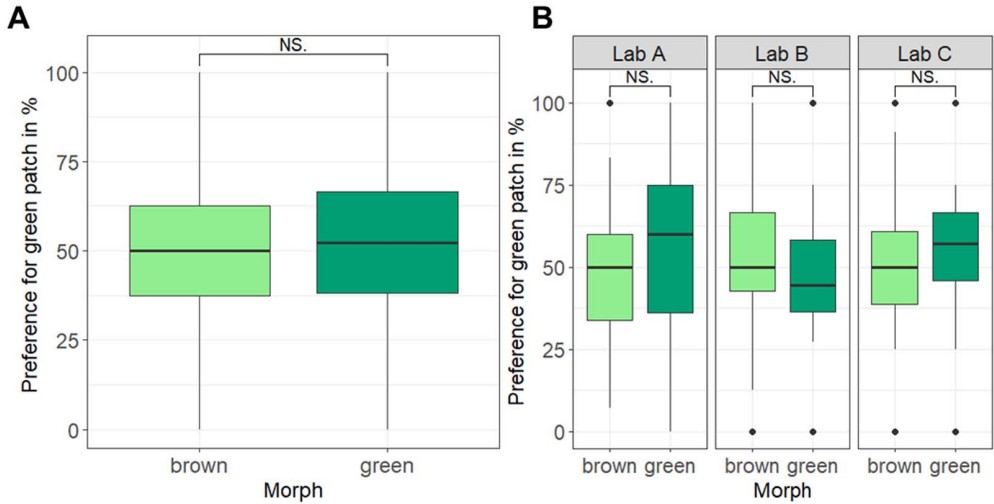

**Fig 4. Preference for the green background out of the two presented backgrounds (green and brown) in the different color morphs of the meadow grasshopper *P. parallelus* (A) across all laboratories and (B) within each laboratory.** Plotted is the preference in percent to sit on the green background, calculated from all observations (8,784 positions across 185 individual grasshoppers). Data are presented as box plots showing medians, 25% and 75% percentiles (lower and upper box), and 5% and 95% percentiles (lower and upper line). Statistics: Wilcoxon signed-rank test, two-sided. Model results from the GLMM on the transformed data show the same direction of significant differences (see Table 2). Raw data and code needed to reproduce this Figure can be found in https://zenodo.org/records/14002690, the summary data presented in the Figure is listed in the S5A and S5B Table.

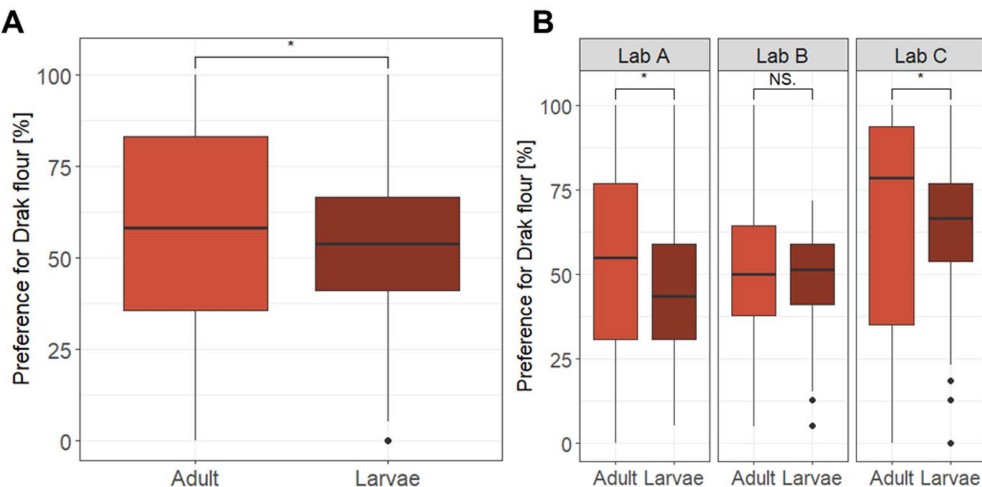

**Fig 5. Effect of treatment (life stage larvae vs. adult) on the flour preference in the red flour beetle *T. castaneum* (A) across laboratories and (B) within laboratories.** Data are presented as boxplots showing medians, 25% and 75% percentiles (lower and upper box), and 5% and 95% percentiles (lower and upper line). Statistics: Wilcoxon signed-rank test, two-sided, on the untransformed data *$p < 0.05$. Model results from the GLMM on the transformed data show the same direction of significant differences (see Table 2). Raw data and code needed to reproduce this Figure can be found in https://zenodo.org/records/14002690, the summary data presented in the Figure is listed in the in S6A and B Table.

adult beetles (Lab A: $W = 3914.5$, $p = 0.03$; Lab C: $W = 8,494$, $p = 0.02$; see Fig 5B). Out of these two cases, the confidence intervals of only one replicate (Lab A) covered the overall effect size, while the other lab (Lab C) observed a larger treatment effect than the overall effect (see Fig 3D). The third laboratory did not replicate significant group differences

(Lab B: $W$ = 7699.5, $p$ = 0.30). Consistent with these findings, the significant 'treatment-by-laboratory' interaction (LRT: $\chi^2(2)$ = 10.51, $p$ = 0.005) echoed the significant differences among laboratories.

### 3.2. Consistency and accuracy of the treatment effects across replicate experiments

When replicating a total of four outcome measures within three unique experimental set-ups, we found overall statistically significant treatment effects in three out of the four outcomes (see Table 3). For the fourth outcome measure, the substrate choice in the grasshopper *P. parallelus*, the treatment effect was neither significant across nor within laboratories (see Table 3). Across all findings, we found the direction of the overall treatment effect ($P_{sig}$) reproduced in 83% of cases (10 out of 12 replicates), while in 58% of cases, both the effect size and significance were accurately predicted ($P_{acc}$; 7 out of 12; see Table 3).

Since the overall treatment effect for the grasshopper experiment did not reproduce the treatment effect of the original study [40], we focused the further reproducibility assessment on the two experiments that successfully replicated their original study effects. For these three outcomes, we were able to reproduce the overall statistical effect ($P_{sig}$) in 77% of the replicates (seven out of nine). But in none of the outcomes did we have perfect reproducibility across all three laboratories. Whenever the overall significant treatment was not reproduced, it was always in the same location (Lab B; see Table 3). The replicate experiments covered the overall effect size of the three significant outcome measures only in 55% of the time ($P_{cov}$; five out of nine cases). In three cases, the effect was overestimated, in one replicate it was underestimated. When evaluating how well the replicate experiments were able to accurately predict both the overall significant treatment effect and also estimate the effect size ($P_{acc}$) for outcome measures, we found this true in 44% (four out of nine outcome measures).

When comparing the reproducibility between the individual outcomes, the distance moved in *A. rosae*, an experiment with automated data collection, had the highest reproducibility. Here all three laboratories reproduced the overall significant treatment effect ($P_{sig}$), and two out of three laboratories also covered the overall effect size as well as predicted the

**Table 3. Overview of all outcome measures, the type of data collection (manual vs. automated), the direction of the overall statistical significance of treatment effects, as well as the respective reproducibility of the replicates.** Highlighted in gray are deviating replicates, that either do not show the same statistical significance ($P_{sig}$), whose $CI_{95}$ do not cover the overall effect size ($P_{cov}$) or which did not accurately predict the statistical significance and did not cover the overall effect size ($P_{acc}$). "Origin" behind a location name marks the respective laboratory that had designed the experiment.

| Experiment | Outcome | Data collection | Overall statistical significance of treatment effect | Replicate experiment (laboratory) | $P_{sig}$ | $P_{cov}$ | $P_{acc}$ | |
|---|---|---|---|---|---|---|---|---|
| *Athalia rosae* | PCI duration | Manual | Significant | Lab A (origin) | √ | X | X | 55% (5/9) |
| | | | | Lab B | X | √ | X | |
| | | | | Lab C | √ | √ | √ | |
| *Athalia rosae* | Distance moved | Automated | Significant | Lab A (origin) | √ | √ | √ | 77% (7/9) |
| | | | | Lab B | √ | X | X | |
| | | | | Lab C | √ | √ | √ | |
| *Pseudochorthippus parallelus* | Substrate choice | Manual | No treatment effect | Lab A | √ | √ | √ | 100% (9/9) |
| | | | | Lab B (origin) | √ | √ | √ | |
| | | | | Lab C | √ | √ | √ | |
| *Tribolium castaneum* | Niche choice | Manual | Significant | Lab A | √ | √ | √ | 44% (4/9) |
| | | | | Lab B | X | X | X | |
| | | | | Lab C (origin) | √ | X | X | |
| | | | | | 83% (10/12) | 66% (8/12) | 58% (7/12) | |

treatment effect ($P_{acc}$; see Table 3). The niche choice of *T. castaneum* (manual data collection) showed the lowest reproducibility, with none of the measures ($P_{sig}$, $P_{cov}$, or $P_{acc}$) being unanimous across all three laboratories.

In addition, the laboratory that originally designed and conducted the experiment did not necessarily cover the overall effect best. More specifically, we found that in one case the inexperienced laboratory replicated the results equally well as the experienced laboratory (*A. rosae*, distance moved). In two cases (*T. castaneum* and the PCI duration of *A. rosae*), did the inexperienced locations even achieve better reproducibility than the laboratory that had developed the experiment.

## 4. Discussion

The reproducibility of studies involving insect species is an underexplored area. Ecological and evolutionary processes often operate over large spatial and temporal scales, making a re-collection of appropriate data difficult and in some cases impossible. Consequently, systematic investigations of reproducibility are scarce in eco-evolutionary studies, rendering it even more important to approach the issue experimentally. Our study aimed to fill this gap by conducting a multi-laboratory study on the reproducibility of three behavioral ecological experiments. Within this framework, we compared the consistency and accuracy with which the replicate experiments predicted the overall treatment effects, both in terms of statistical significance and effect size.

Across our three experiments, we documented an overall prevalence rate of irreproducibility of 17–42%. The 42% thereby reflect the proportion of irreproducibility according to the strictest criterion ($P_{acc}$), namely the correct estimation of the effect size in combination with the replication of the significance of the overall treatment effect, while only 17% of the results did not reproduce the same direction of significance ($P_{sig}$). With respect to our third experiment (*P. parallelus*), none of the laboratories were able to reproduce the previously described significant treatment effect.

With these findings, this study documents challenges in achieving reproducibility of ecological studies on insect behavior. In comparison to other systematic replication studies [3,6,7], however, we observed higher reproducibility rates, suggesting that while reproducibility issues do exist in insect studies, they might be less pronounced than in other areas of science.

### 4.1. Causes of poor reproducibility

**4.1.1. Standardization and reproducibility.** According to the previously postulated "standardization fallacy" [34,35], different laboratories might produce increasingly idiosyncratic results as standardization within laboratories becomes more rigorous (see also [1]). In the present study, potential effects of "over-standardization" became evident on three levels: First, we found significant "treatment-by-laboratory" interactions for *A. rosae* and *T. castaneum* experiments, indicating that the detected treatment might have been idiosyncratic to the specific laboratory, which would match the findings of Crabbe and colleagues (1999). This is even more surprising, as protocols were harmonized across laboratories, thereby reaching a higher standard than is usually reached in replicate studies. In fact, a recent study on pharmacological effects in mice clearly demonstrated that harmonization of protocols across laboratories reduced between-laboratory variation substantially compared to a situation, where each laboratory uses its own local protocol [70]. In contrast, a recent analysis of coordinated distributed experiments in ecology suggests that reducing methodological heterogeneity across sites does not consistently reduce variation in observed effect sizes, possibly due to higher intrinsic biological variability among locations that predominates over methodological variance [71]. Thus, the effectiveness of standardization in minimizing between-laboratory variability may depend upon the inherent biological heterogeneity characteristic of the study system.

Second, we covered different levels of standardization by including three species that inherently varied in their degrees of genetic diversity and habituation to the laboratory: *P. parallelus* was the least standardized population, with unknown life histories and no habituation to laboratory conditions. In contrast, *T. castaneum* represented the most standardized model, with all individuals stemming from a homogenous, laboratory-adapted population that had been controlled for age and life history. *A. rosae* fell in between, descending from a laboratory stock population that was annually augmented with

wild-caught individuals, but individuals used in the experiment were already from the 12th generation of purely laboratory-reared cohorts. Interestingly, the results of the *A. rosae* and *T. castaneum* experiments were not as consistent across our three laboratories as might be expected by such a comparatively high degree of standardization. By contrast, when focusing on the results of our study and neglecting the comparison with the original study, the results of the *P. parallelus* experiment were the most consistent across replicates. This might indicate that greater heterogeneity of the individuals within a study species could potentially benefit the representativeness of the study population and hence lead to better reproducibility across studies [29]. Similarly, an ecological study on grass grown in microcosms showed that a controlled systematic increase in genetic variation reduced variation among laboratories [72]. Apart from that, differences in reproducibility might also stem from variability in parental environmental conditions. While all *A. rosae* individuals were taken from one generation and population, *T. castaneum* individuals came from different generations, and wild-caught *P. parallelus* very likely also belonged to different cohorts. Finally, the lack of a significant treatment effect in the *P. parallelus* experiments could also have made it easier to achieve reproducibility.

Third, in regards to reproducing results from the original studies, *A. rosae* and *T. castaneum* experiments performed markedly better than the *P. parallelus* experiment, as none of our laboratories found any substrate preference as a function of color morph. Thus, although the reproducibility seen within our multi-laboratory approach may be considered high for this experiment, we failed to reproduce the effect seen in the original study [40]. Notably, out of the three experiments, the experimental set-up of the *P. parallelus* experiment deviated the most from the design of the original study. For instance, our experiment differed in terms of cage size, substrate design, and age structure, potentially explaining the observed discrepancies. At the same time, however, such conflicting findings highlight problems arising from over-standardization within experiments, as this makes it even harder to reproduce findings across the variation that inevitably exists between experiments and laboratories [29,30].

**4.1.2. Specific experimental variables and reproducibility.** In rodent studies, significant research has been conducted to identify and rank sources of experimental variation. Along these lines, it has been repeatedly highlighted that one of the most confounding factors is the experimenter [73–75]. Precisely what differentiates experimenters remains unknown, but there is some evidence that aspects such as the sex [76,77] or the familiarity and training of the experimenter play a decisive role [78]. Until now, however, it is unclear whether similar patterns also hold for insect studies. Theoretically, the effect of the experimenter depends on an organism's sensitivity to environmental cues. Zebrafish models, for instance, appear to be more resilient to experimenter effects compared to rodents, likely because they are less exposed to experimenter-specific pheromones and odor [79]. In the present study, we standardized the experience levels of the lead experimenters, ensuring that none had prior experience with the specific study organisms or behavioral assays. In two of the laboratories, however, they were assisted by researchers experienced with insect species and bioassays related to their laboratory. Adopting this approach ensured similar conditions across laboratories, thereby reducing but not excluding the risk of introducing any experimenter effects.

Reproducibility also varied with the levels of manual handling and scoring required: The variables measured by automated tracking software, i.e., the distance moved in the sawfly larvae, yielded the most consistent results across laboratories, while the behavior requiring manual handling in the same experiment, i.e., triggering PCI, showed substantially greater variation. In line with these findings, automated systems have been shown to yield better reproducible results across laboratories in comparison to manually conducted tests in rodents [80–82]. Likewise, it has been argued that computer algorithms, once programmed and trained, lead to more consistent and unbiased measurements [83,84], suggesting that the absence of human interference is a prominent advantage (cf. [85]).

Furthermore, reproducibility varied by trait: effects of starvation on the PCI duration in *A. rosae* larvae and substrate preferences of *T. castaneum* showed high variation between laboratories. Similarly, research on rodents found highly reproducible strain differences for ethanol preference and locomotor activity, while strain differences for anxiety-like behavior strongly depended on the local conditions [86]. Most likely, such findings indicate that processes underlying the

different constructs might be more or less sensitive to environmental fluctuations, hence yielding more or less idiosyncratic findings.

Lastly, species-specific characteristics might also contribute to differences in reproducibility. In this respect, the *T. castaneum* experiment highlights how age can impact reproducibility. Whereas adult behavior was consistent with the original study, larvae here showed a preference for secretion-reduced flour, unlike the original study [41]. This difference might be explained by our use of older larvae (17 vs. 12 days), with the older aged larvae likely reaching a different instar, closer to pupation. While younger larvae feed constantly [87] and likely depend on external immunity provided by the antimicrobial secretions in the flour, old larvae nearing their pre-pupal phase might prefer a niche with lower densities to avoid dangers of cannibalism. Here, the secretion-reduced flour could act as a signal of lower density of conspecifics [88]. Similarly, some *A. rosae* larvae may have been near molting in certain laboratories. During the molting period, larvae do not feed, and thus starvation may not have had a strong effect on them. Standardizing age in larvae might, however, be particularly challenging when larval stages are hard to discriminate.

### 4.1.3. Sample size, statistical significance, and reproducibility.

Larger sample sizes are known to increase the statistical power of a study, i.e., the probability of rejecting a false null hypothesis [89,90]. It has thus been argued that poor reproducibility of rodent studies is at least partly due to a lack of sufficiently powered experiments [91]. In invertebrate studies, sample sizes are typically considerably higher and might include hundreds or even thousands of animals [92,93]. The sample sizes included in our experiments (see Table 1 and S4–S7 Tables) were well in the range to be considered sufficiently powered [91]. Despite this, however, we still observed limited reproducibility of some findings, indicating that increasing the power is one important step toward improvement, but certainly not the only one.

Apart from that, the reproducibility crisis has also led to growing concerns about the sole use of *p*-values and statistical significance for reporting findings. In particular, it has been argued that degrading *p*-values into significant and non-significant findings contributes to making studies irreproducible, or to making them *seem* irreproducible [94]. In the present study, we used three different measures to compare the reproducibility of results across replicate experiments. These measures increased in their rigor from evaluating reproducibility based solely on the significance of *p*-values, to the replication of the overall effect sizes and, finally, to the combination of both conditions. In contrast to *p*-values, effect sizes are independent of sample size. Adding effect sizes and their confidence intervals to publications has therefore been proposed to contribute to a more nuanced interpretation of experimental data [95,96]. We were able to reproduce the overall statistical significance of the treatment effect in 83% of the replicate experiments; however, reproducing the overall effect size was successful in only 66%, while both criteria were achieved in 58% of cases. Such a discrepancy between replicating statistical significance and effect size has been observed in other fields, too. For example, in cancer biology, 79% of replication studies had the same statistical significance, while only 18% reproduced the original effect size [7]. Whether effect sizes or statistical significance are easier to reproduce appears to be field-specific (in psychology: 47% versus 36% [6]; in economics: 61% versus 67% [10]).

### 4.2. Toward better reproducibility in insect studies

Identifying sources of poor reproducibility represents only the first step toward improvement. It is just as important to foster ways out of the crisis. In this respect, a number of strategies have already been developed that address methodological shortcomings or aim to improve the overall publication culture [23,24,26]. Although most of these strategies have been tailored towards rodent studies, they can be easily broadened to also encompass insect experiments. For instance, identifying additional reliable outcome measures in insect behavior could enhance the robustness of insect studies. This approach mirrors successful techniques used in rodent research, where the use of computer algorithms has expanded the repertoire of commonly used behavioral paradigms to include newly established measures, such as specific movement patterns [97–99]. Likewise, to minimize potential experimenter effects, several methodological strategies, such as blinding or the use of automated test systems, could be implemented more systematically [78,85,100,101]. Also, replicating studies

independently and increasing sample sizes per single experiment would be more easily possible in insect studies, as invertebrates are comparatively easy to keep and breed and experiments are less constrained by ethical considerations and regulatory restrictions. Furthermore, to address the problem of over-standardization, systematic heterogenization of experimental conditions within laboratories has been proposed as a tool to deliberately incorporate known sources of biological variation into the experimental design (e.g., [1,27–29,102]). According to this idea, the introduction of variation on a systematic and controlled basis predicts increased external validity and hence improved reproducibility [30]. For example, one could imagine introducing biological variation by using insect populations from multiple geographic locations or from different cohorts or generations. For some insects, in particular *Drosophila*, also different levels of genetic homogenization can be included (DGRP lines; [103]). Regarding experimental conditions, studies could be conducted across different rooms, spread across weeks or seasons [66], or, depending on the biology of the model species, could also be realized at different times of the day [104]. Lastly, an implementation of open research practices throughout the research cycle is strongly advocated, independent of the research area [105,106].

## 5. Conclusions

With the present study, we aimed to systematically explore the reproducibility of ecological studies on insect behavior across independent replicate experiments. By means of a multi-laboratory approach, we uncovered difficulties in reproducing the overall estimate of the effect size as well as the direction of significance, documenting that reproducibility problems might also exist in insect studies. However, parts of the results are also encouraging, since reproducibility in our experiments was larger than in replication studies from other fields. This way, we wish to raise awareness for the topic and encourage the implementation of potential improvement strategies. We believe that addressing the reproducibility crisis requires a comprehensive solution involving methodological improvements, better experimental design, and a collaborative effort across research communities. Specifically, adopting open research practices and introducing systematic variation through multi-laboratory or heterogenized designs as well as implementing preregistrations may further enhance reproducibility in insect studies.

## Supporting information

**S1 Fig. Arrangement of Petri dishes for the *A. rosae* experiment with starvation or control treatment.** Photo credit: Maximilian Schurig.
(TIF)

**S2 Fig. Top-down view of a single cage for the *P. parallelus* experiment.** Photo credit: Maximilian Schurig.
(TIF)

**S3 Fig. Setup of a single Petri dish containing (A) three female virgin adults and (B) three larvae for the *T. castaneum* experiment.** Photo was taken 24 h after the start of the experiment. Photo credit: Maximilian Schurig.
(TIF)

**S4 Fig. Substrate preference in *P. parallelus* morphs across all labs, split by sex.** Data are presented as boxplots showing medians, 25% and 75% percentiles, and 5% and 95% percentiles. Statistics: Wilcoxon signed-rank test, two-sided, on the untransformed data $*p < 0.05$, $**p < 0.01$, $***p \leq 0.001$. The data and code needed to reproduce this Figure can be found in https://zenodo.org/records/14002690. The data summarized in the Figures can be found in S7 Table.
(TIF)

**S5 Fig. Choice of *P. parallelus* across all laboratories for sitting on one of the substrate patches (brown or green), or on neither patch (instead sitting, e.g., on the walls or lid of the cage).** In 15.1% of those observations (1,329 instances), grasshoppers were sitting on the floor and could thus be assigned to one of the two patch colors. In the

remaining 84.9% of observations (7,455 instances), grasshoppers were sitting on the cage walls, under the cage lids, or on the bundles of grass. Data are presented as boxplots showing medians, 25% and 75% percentiles, and 5% and 95% percentiles. The data and code needed to reproduce this Figure can be found in https://zenodo.org/records/14002690. The data summarized in the Figures can be found in S8 Table.
(TIF)

**S1 Table. Details on housing conditions, animals, preparation of materials and setup, experimental phase, and experimenter-specific characteristics for the Athalia experiment for each laboratory.**
(DOCX)

**S2 Table. Details on housing conditions, animals, preparation of materials and setup, experimental phase, and experimenter-specific characteristics for the Pseudochorthippus experiment for each laboratory.**
(DOCX)

**S3 Table. Details on housing conditions, animals, preparation of materials and setup, experimental phase, and experimenter-specific characteristics for the *Tribolium* experiment for each laboratory.**
(DOCX)

**S4 Table. (A) Descriptive Statistics of the outcome measure "PCI duration" [sec] in the *Athalia* experiment for each group across all labs. (B) Descriptive Statistics of the outcome measure "PCI duration" [sec] in the *Athalia* experiment within each lab and group. (C) Descriptive Statistics of the outcome measure "distance moved" [cm] in the Athalia experiment for each group across all labs. (D) Descriptive Statistics of the outcome measure "distance moved" [cm] in the Athalia experiment within each lab and group.**
(DOCX)

**S5 Table. (A) Descriptive statistics of the outcome measure "substrate choice" as percent of individuals on green patch [%] in the *Pseudochorthippus* experiment for each morph type across all labs. (B) Descriptive statistics of the outcome measure "substrate choice" as percent of individuals on green patch [%] in the *Pseudochorthippus* experiment within each lab and morph type.**
(DOCX)

**S6 Table. (A) Descriptive statistics of the outcome measure "niche choice" as percent of individuals in drak-flour [%] in the *Tribolium* experiment for each group across all labs. (B) Descriptive statistics of the outcome measure "niche choice" as percent of individuals in drak-flour [%] in the *Tribolium* experiment within each lab and group.**
(DOCX)

**S7 Table. Descriptive statistics of the outcome measure "substrate choice" as percent of individuals on green patch [%] in the *Pseudochorthippus* experiment within each lab, sex, and morph type.**
(DOCX)

**S8 Table. Descriptive statistics of the recorded location [%] across all individuals in the *Pseudochorthippus* experiment across all labs.**
(DOCX)

## Author contributions

**Conceptualization:** Joachim Kurtz, Caroline Müller, Holger Schielzeth, Vanessa T. von Kortzfleisch, S. Helene Richter.

**Data curation:** Carolin Mundinger, Nora K. E. Schulz, Pragya Singh, Vanessa T. von Kortzfleisch.

**Formal analysis:** Carolin Mundinger.

**Funding acquisition:** Joachim Kurtz, Caroline Müller, Holger Schielzeth, S. Helene Richter.

**Investigation:** Nora K. E. Schulz, Pragya Singh, Steven Janz, Maximilian Schurig, Jacob Seidemann.

**Methodology:** Nora K. E. Schulz, Pragya Singh, Steven Janz, Maximilian Schurig, Jacob Seidemann, Vanessa T. von Kortzfleisch.

**Project administration:** Vanessa T. von Kortzfleisch, S. Helene Richter.

**Resources:** Joachim Kurtz, Caroline Müller, Holger Schielzeth.

**Supervision:** Nora K. E. Schulz, Pragya Singh, Joachim Kurtz, Caroline Müller, Holger Schielzeth, S. Helene Richter.

**Visualization:** Carolin Mundinger.

**Writing – original draft:** Carolin Mundinger, Nora K. E. Schulz, Pragya Singh, S. Helene Richter.

**Writing – review & editing:** Carolin Mundinger, Steven Janz, Maximilian Schurig, Jacob Seidemann, Joachim Kurtz, Caroline Müller, Holger Schielzeth, Vanessa T. von Kortzfleisch, S. Helene Richter.

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
