## [Editor Report · Decision Letter 0]

10 Jan 2025

Dear Helene,

Thank you for submitting your manuscript entitled "Three species, three labs, three experiments: testing the reproducibility of ecological studies on insect behaviour in a multi-laboratory setting" for consideration as a Meta-Research Article by PLOS Biology. Please accept my apologies for the extreme delay incurred during the two-week journal office closure over the holiday period.

Your manuscript has now been evaluated by the PLOS Biology editorial staff, and I am writing to let you know that we would like to send your submission out for external peer review. I should note that we weren't able to secure advice from an appropriate expert, but after discussion among the team we're happy to see what reviewers think of it.

Once your full submission is complete, your paper will undergo a series of checks in preparation for peer review. After your manuscript has passed the checks it will be sent out for review. To provide the metadata for your submission, please Login to Editorial Manager (https://www.editorialmanager.com/pbiology) within two working days, i.e. by Jan 14 2025 11:59PM.

Kind regards,

Roli

Roland Roberts, PhD

Senior Editor

PLOS Biology

rroberts@plos.org

---

## [Decision Letter · Decision Letter 1]

14 Mar 2025

Dear Helene,

Thank you for your patience while your manuscript "Three species, three labs, three experiments: testing the reproducibility of ecological studies on insect behaviour in a multi-laboratory setting" was peer-reviewed at PLOS Biology. It has now been evaluated by the PLOS Biology editors, an Academic Editor with relevant expertise, and by two independent reviewers. A third reviewer was also recruited, but they were not able to return their comments in a timely fashion. Please accept my apologies for the extreme delay incurred.

You'll see that reviewer #1 is very positive, and only has textual and/or presentational requests. Reviewer #2 similarly is positive, and again only has minor requests for clarification, etc.

Based on the reviews, we are likely to accept this manuscript for publication, provided you satisfactorily address the remaining points raised by the reviewers and the following data and other policy-related requests.

IMPORTANT - please attend to the following:

a) Please attend to the requests from the reviewers. The Academic Editor noted: "In particular, I agree with reviewer 2's suggestion to add more information about the diet and prenatal conditions of the organisms."

b) Please could you change your Title to "Testing the reproducibility of ecological studies on insect behavior in a multi-laboratory setting identifies opportunities for improving experimental rigor" (we try to avoid punctuation in Titles, we like to include some of the findings, and the specific study design will be immediately evident in the Abstract)

c) Please address my Data Policy requests below; specifically, we need you to supply the numerical values underlying Figs 2ABCD, 3ABCD, 4AB, 5AB, S4, S5, either as a supplementary data file or as a permanent DOI’d deposition. I note that you already have an associated Zenodo deposition, but this is not currently accessible; please either make it accessible or send me a reviewer login so that I can check compliance.

d) Please cite the location of the data and code clearly in all relevant main and supplementary Figure legends, e.g. “The data and code needed to reproduce this Figure can be found in "https://zenodo.org/records/14002691"

e) Please make any custom code available, either as a supplementary file or as part of your Zenodo data deposition (I'm assuming it's already in there?).

We expect to receive your revised manuscript within two weeks.

*Published Peer Review History*

*Press*

Sincerely,

Roli

Roland Roberts, PhD

Senior Editor

rroberts@plos.org

PLOS Biology

DATA POLICY:

Regardless of the method selected, please ensure that you provide the individual numerical values that underlie the summary data displayed in the following figure panels as they are essential for readers to assess your analysis and to reproduce it: Figs 2ABCD, 3ABCD, 4AB, 5AB, S4, S5. NOTE: the numerical data provided should include all replicates AND the way in which the plotted mean and errors were derived (it should not present only the mean/average values).

CODE POLICY

REVIEWERS' COMMENTS:

Reviewer #1:

This study is an important addition to the growing body of work devoted to describing and attempting to explain variability in results among empirical studies. These researchers demonstrate, in studies of three very different species of insects, that observed behavioral responses to apparently identical experiments sometimes differed substantially when conducted in different labs. This is similar to results that have been observed in a number of sets of laboratory rodent studies. This current study is useful in that it demonstrates this phenomenon in a different group of organisms, and because it brings the work to the attention of a different set of researchers - animal behavior researchers more closely tied to the disciplines of evolutionary biology and ecology.

The study appears well designed and well executed, and the data analyses appear reasonable.

I would be interested to read the authors' thoughts on the implications of their work for understanding the substantial heterogeneity that has been demonstrated to exist among published statistical effects in ecology and evolutionary biology (for instance, by Senior et al. 2016. Ecology, 97:3293-3299).

I have arranged my other comments by line number.

66 - "Attempts to replicate core scientific finding have produced mixed to rather discouraging results (e.g., Ioannidis, 2005; Freedman et al., 2015; Baker, 2016),"

The "e.g.," here is misleading, because it implies that you are listing examples of studies that attempted to replicate scientific findings. However, the studies you cited do not actually attempt replications, themselves. The Baker paper surveyed researchers about the practices and experiences, and the Ioannidis paper mostly presents a model and an argument. I don't know the Freedman et al. paper.

So, the solution would be to either cite papers that actually assessed replicability, or to delete "e.g.,"

Also, I think the term 'core' overstates the case here. Are these findings that failed to replicate actually at the core of science such that their failure to replicate brings down the entire edifice?

69 - "70% of researchers reported,"

I suggest editing to something like "70% of researchers who responded to a survey reported…" since these surveys are not representative

75 - (and throughout) - I encourage you to reconsider your widespread us of the term 'reproducibility' and to instead use 'replicability'.

There is a strong trend in the growing metascience movement to converge on the use of the terms according to these definitions:

(taken from https://www.ncbi.nlm.nih.gov/books/NBK547546/)

"… reproducibility is obtaining consistent results using the same input data; computational steps, methods, and code; and conditions of analysis. This definition is synonymous with "computational reproducibility," ….

Replicability is obtaining consistent results across studies aimed at answering the same scientific question, each of which has obtained its own data."

86 - do you mean "guides" (plural)?

111 - You may wisht to consult and incorporate results in this paper:

Milcu et al. 2018 https://doi.org/10.1038/s41559-017-0434-x

This paper examines the issue in an ecological study using plant microcosms

224,227 - The treatment is confusing. Four hours without food, and then 'after three hours of treatment'. So, does this mean 4+3 = 7 hours of starvation, or something else?

OK, I see that this is clarified later in the paragraph, but it would be useful to clarify it immediately.

344: the zenodo files are not publicly accessible, and so I cannot assess them

389: which type of effect size is this?

422: "did not detect any differences… p = 0.14"

I encourage you to moderate you wording here. The means differed, it's just that the p-value did not cross your arbitrary threshold. I suggest "did not detect a statistically significant difference".

I notice that you cite Amrhein et al. in the discussion - their perspective is relevant here.

Fig 3. Please identify the type of effect size.

Also, it seems that it could be valuable to also indicate on these graphs the effect sizes produced by the original published studies being replicated here.

471: I think a word may be missing after "a"

479: "could not be reproduced" implies an effort to get the same result. Wouldn't it be more accurate to say "did not reproduce" since this does not imply an effort to get a particular result? (or actually, "did not replicate", see earlier comment)

Figure 5. The Y-axis is labeled as "preference for dark flour", but no where in the paper do you discuss "dark flour"

553: Baker 2016 was not a direct study of reproducibility / replication. It was a survey of researchers experiences with replication

564: "indicating that the detected treatment effects were indeed idiosyncratic to the specific laboratory"

I think this claim is somewhat misleading. The detected treatment effects in this case were idiosyncratic to the different implementations of the experiment, but based on the information available, we cannot be certain that the differences are attributable to the labs per se. For instance, it is possible that if you replicated this study in these same labs again, the results would differ among labs, but not with the same pattern.

570: However, interestingly, a recent study in ecology suggests that standardizing methods does not necessarily reduce heterogeneity in results in that discipline.

Bebout, J. and Fox, J.W. (2024), Coordinated distributed experiments in ecology do not consistently reduce heterogeneity in effect size. Oikos, 2024: e10722. https://doi.org/10.1111/oik.10722

It might be interesting to consider why the Bebout and Fox study come to a different conclusion. I suspect it may be due to the dramatically higher true biological heterogeneity among locations in many of the systems they considered

599: some prior work, for instance in psychology, has found evidence that heterogeneity among results is higher when the true effect size is larger. It would be interesting to consider the applicability of this idea to your findings. See, for instance:

Klein Richard A, Vianello Michelangelo, Hasselman Fred, Adams Byron G, Adams Reginald B, Alper Sinan, Aveyard Mark, et al. Many Labs 2: Investigating Variation in Replicability Across Samples and Settings. Advances in Methods and Practices in Psychological Science. 2018;1(4):443-90. https://doi.org/10.1177/2515245918810225.

621: not directly compared statistically, so maybe 'substantially' instead of 'significantly'?

671: as worded here, it is not clear that you are discussing statistical *significance*, as effect sizes could also be described as "overall statistical treatment". Please be explicit about when you are discussing significance testing.

Reviewer #2:

The topic of reproducibility in science, the topic of this ms, is extremely important, and this is a solid addition to this literature. Experiments similar to what was done here have been performed for other taxa, including the landmark mouse study by Crabbe et al., but the authors claim this is the first such study for insects, and this reviewer has not seen otherwise.

The study is very comprehensive, and well executed, with a now standard 3 experimental site experimental design. In addition, the experiment involves 3 different experiments, each conducted at the 3 sites, with experimental replicates also included at each site. The findings parallel the rodent studies of this type. This is both predictable and at the same time novel, in that one might have not expected that some of the same "handling type" factors identified as being important for vertebrates would also be important for insects, given the assumption that insects are in some way less sensitive to such factors. That they are is cause for re-evaluation of experimental protocols, a provocative finding sure to spark further study and consideration.

The Introduction is excellent, identifying the important issues related to the reproducibility crisis, and how the present study relates to them. The rational for the species chosen and experiments conducted is cogent and thoughtful. The study was well designed.

The Discussion also is an excellent and thoughtful treatment of the issues and how the results of this study fit.

I have the following relatively minor comments for possible revision.

Methods. As I recall the Crabbe et al study made a concerted effort to standardize the diets across the lab, including the sourcing of the diet components. I did not find such information in the present study. Was it done? And if not, why not?

The red flour beetle results mirror the earlier rodent results, with variation in outcome despite genetic homogeneity due to lab-based breeding. That the variation was not attenuated relative to the wild-caught species is noteworthy. It would be good if the authors could comment on this a bit more in the Discussion.

One possible source of variation could be the condition of the individuals based on prenatal environmental influences. Perhaps individuals came from the same populations but were of different generation? This would provide a source of variation not included in the experimental paradigm. This could be minimized by sample size, unless there are cohort differences. It would be good for the authors to comment on this possibility, even if they do not consider it to be a possible source of variation. The statistical models do not seem to consider the possibility of intrinsic differences in physiology and behavior.

In some cases animals were shipped from a common source. Could there have been any variation in shipping conditions/duration? Train transport (Exp 2) unlikely but through the mail service in other experiments?

Several figure captions (e.g., Fig. 2, 4, etc.) do not explain the figure well, i.e., what the horizontal line in the bar represents, the dots—mean/median/CI? (Some or all of the data? Summarized in what way?). Other figures do a good job of this.

---

## [Editor Report · Decision Letter 2]

25 Mar 2025

Dear Helene,

Thank you for the submission of your revised Meta-Research Article "Testing the reproducibility of ecological studies on insect behaviour in a multi-laboratory setting identifies opportunities for improving experimental rigour" for publication in PLOS Biology. On behalf of my colleagues and the Academic Editor, Ingrid Fetter-Pruneda, I'm pleased to say that we can in principle accept your manuscript for publication, provided you address any remaining formatting and reporting issues. These will be detailed in an email you should receive within 2-3 business days from our colleagues in the journal operations team; no action is required from you until then. Please note that we will not be able to formally accept your manuscript and schedule it for publication until you have completed any requested changes.

Sincerely, 

Roli

Senior Editor

PLOS Biology

rroberts@plos.org